# Advancing Precision in Neuro-Oncology with Intraoperative Imaging and Fluorescence Guidance: A Narrative Review

**DOI:** 10.3390/biomedicines13102550

**Published:** 2025-10-20

**Authors:** Małgorzata Podstawka, Anna Dębska, Bartosz Szmyd, Karol Zaczkowski, Michał Piotrowski, Ernest J. Bobeff, Paweł Ratajczyk, Dariusz J. Jaskólski, Karol Wiśniewski

**Affiliations:** 1Department of Neurosurgery and Neurooncology, Barlicki University Hospital, Medical University of Lodz, 90-153 Lodz, Poland; anna.debska2@stud.umed.lodz.pl (A.D.); bartosz.szmyd@umed.lodz.pl (B.S.); karol.zaczkowski@stud.umed.lodz.pl (K.Z.); michal.piotrowski@stud.umed.lodz.pl (M.P.); ernest.bobeff@umed.lodz.pl (E.J.B.); dariusz.jaskólski@umed.lodz.pl (D.J.J.); karol.wisniewski@umed.lodz.pl (K.W.); 2Department of Sleep Medicine and Metabolic Disorders, Medical University of Lodz, 90-419 Lodz, Poland; 3Department of Anesthesiology and Intensive Therapy, Medical University of Lodz, 90-153 Lodz, Poland; pawel.ratajczyk@umed.lodz.pl

**Keywords:** glioma, resection, glioblastoma, virtual MR, intraoperative MR

## Abstract

Malignant gliomas remain among the most formidable challenges in neuro-oncology, given their high morbidity and rising incidence worldwide. Surgical resection represents the cornerstone of treatment, typically followed by adjuvant radiotherapy and chemotherapy. Achieving maximal safe resection, however, requires advanced intraoperative guidance. A range of adjuncts are currently employed, including 5-aminolevulinic acid (5-ALA), intraoperative ultrasound, computed tomography (iCT), and intraoperative magnetic resonance imaging (iMRI). More recently, an emerging technique—virtual MRI (vMRI)—has been developed, fusing intraoperative CT with preoperative high-resolution MRI to provide real-time, MRI-like updates of brain anatomy. Beyond imaging, tumour removal itself induces reorganization of eloquent brain networks, underscoring the critical need for precision tools that balance oncological control with preservation of neurological function. In this narrative review, we highlight and synthesize the evolving armamentarium of intraoperative technologies shaping the future of precision neuro-oncology.

## 1. Introduction

The most common primary central nervous system (CNS) tumours encompass IDH-wildtype glioblastoma (GB), astrocytomas IDH mutant (WHO 2–4), oligodendroglioma, and anaplastic oligodendroglioma (both IDH mutant with 1 p/19 q codeletion). Age has a significant impact on the incidence of these tumours, with the vast majority of cases occurring in people over the age of 40. In 47.9% of study participants, the age at diagnosis of GB was over 65 years; similarly, 46.3% of study participants were aged between 40 and 64 years. There is a huge difference in the prognosis between tumour types—median overall survival rate (OSR) for patients with GB is approximately 15 months, whereas those with astrocytoma IDH mutant grade 2 live up to 15 years [1].

GB is one of the most aggressive malignant tumours and the most common primary malignant brain tumour, accounting for 14.5% of all CNS tumours and 48.6% of malignant brain tumours. The incidence of primary brain tumours has increased in recent decades, particularly in the elderly, with the incidence of GB varying from 3.19 to 4.17 cases per 100,000 person-years depending on the report. The elderly represent a stable population of patients with GB. The incidence of GB is 3.23 per 100,000 person-years and is higher in people over 40 years of age (6.97 per 100,000 person-years), peaking at 75–84 years of age (15.30 per 100,000 person-years). GB is mainly located in the frontal, temporal, and parietal lobes, with less frequent involvement of other structures. The number of detected cases has increased in the last two decades (increased incidence/better diagnostic techniques). The only proven factor leading to the development of a malignant glial tumour is radiotherapy [2].

High-grade IDH-mutated astrocytomas (grades 3 and 4) are less frequent than GB. In Europe, the annual incidence of grade 3 astrocytomas is approximately 0.3 cases per 100,000 population. They account for 4% of all CNS malignancies in population-based registries [3]. According to the Surveillance, Epidemiology, and End Results 21 registries, conducted over a period of 14 years, the incidence of these entities is 4.1 per 100,000. It is more prevalent in individuals over 65 years of age and occurs more frequently in men. They have an increased mortality, accounting for 2.5% of cancer deaths in young adults. Life expectancy at 5 years is less than 5%, with a median survival of 14–18 months with standard treatment. IDH-mutant grade 4 astrocytomas are usually diagnosed in young adults, at a median age of 38 years. This is significantly earlier than GB (median age 50–60 years). The clinical history is usually short (3–6 months) [3].

The symptoms associated with the presence of a brain tumour can be divided into three types. The first is a progressive neurological deficit (68%), associated with damage to a specific brain structure; the most common is a motor deficit (45%). The second is associated with increased intracranial pressure and results in headaches (54%). The third type of symptom is an epileptic seizure (26%), often a partial secondarily generalized seizure [1,4].

Magnetic resonance imaging (MRI) is the gold standard for the preoperative diagnosis of brain tumours. The imaging sequences, including T1, T2, FLAIR, SWI, DWI, and tractography, provide information about tumour cellularity, cerebral edoema, cerebral invasion and white matter tracts. High-grade tumours show strong contrast enhancement and frequent central necrosis, often accompanied by tumour ring enhancement. Functional MRI enables mapping of eloquent cortical centres, their relationship to the tumour, and associated neuroplasticity [5].

The cornerstone of glioma treatment is surgery, followed by radiotherapy and chemotherapy (Stupp protocol) [4,6,7,8]. For most patients with brain tumours, diagnostic or cytoreductive surgery is the first stage of treatment. It is extremely important that it is as complete as possible, whilst not causing any serious neurological dysfunction that would hamper further treatment. The complete resection (gross total resection; GTR) with adjuvant radiotherapy and chemotherapy extends survival to a much longer period than a partial resection. The least effective treatment is radiotherapy and chemotherapy alone [6,7,8]. Meanwhile, complete macroscopic removal of these tumours poses several intraoperative challenges. Intraoperatively, gliomas often appear very similar to the surrounding brain tissue, with no distinct boundaries. Furthermore, they grow aggressively, invading adjacent vessels and eloquent brain regions. For this reason, intraoperative imaging techniques in gliomas are crucial for the safe and complete resection of the tumour. They allow for precise localization of the tumour and delineation of its margins and adjacent structures. To aid resection, surgeons most commonly use intraoperative ultrasound (iUS), computed tomography (iCT), or MRI. Other tools include 5-aminolevulinic acid (5-ALA), which improves intraoperative identification of the tumour by visualizing it under ultraviolet light, and fluorescein, which accumulates in areas of blood–brain barrier disruption, aiding in the visualization of contrast-enhancing lesions [9].

The goal of this study is to provide a comprehensive, evidence-based overview of current intraoperative imaging and visualization techniques used in brain tumour surgery, including iMRI, iCT, iUS, virtual MRI (vMRI), and fluorescence-guided surgery using 5-ALA and fluorescein. The study aims to evaluate the advantages, limitations, mechanisms of action, and levels of clinical evidence supporting each technique, with the objective of informing optimal surgical decision-making and enhancing the safety and extent of tumour resection in neurosurgical oncology. A narrative review was chosen to highlight the most important aspects of the evolving armamentarium of intraoperative technologies shaping the future of precision neurooncology.

## 2. Intraoperative Ultrasound (iUS or USG)

Intraoperative ultrasound (iUS or USG) is an increasingly valuable imaging modality used during brain tumour surgery, providing real-time, cost-effective, and radiation-free guidance for neurosurgeons. Unlike preoperative MRI or CT, which offer only static images, iUS allows continuous, dynamic visualization of the brain during surgery, helping in real time to guide tumour localization, assess resection progress, and detect residual tumour [10]. One of the key advantages of ultrasound is its ability to visualize changes intraoperatively without significantly interrupting the surgical workflow, as it can be quickly repeated throughout the procedure. It is particularly useful in cases where the tumour is located deep within the brain or in eloquent regions, as it enables the surgeon to distinguish between normal and abnormal tissue based on differences in echogenicity [11]. Moreover, iUS helps compensate for brain shift by offering updated anatomical images during surgery, enhancing the accuracy of neuronavigation systems that rely on preoperative data [12].

Although conventional B-mode ultrasound images may initially be difficult to interpret due to their lower resolution compared to MRI, experience and improved image processing have significantly increased their clinical utility. Advanced ultrasound techniques such as contrast-enhanced ultrasound (CEUS), Doppler imaging, and elastography have expanded the functionality of iUS beyond basic structural imaging. For example, CEUS can enhance the delineation of tumour vascularity and margins, especially in high-grade gliomas (HGG), and Doppler modalities enable identification of blood vessels, reducing the risk of intraoperative bleeding [13]. In some settings, navigated 3D ultrasound can be integrated with preoperative MRI to fuse images, offering a hybrid solution that combines the real-time benefits of ultrasound with the anatomical detail of MRI [14].

Compared to intraoperative MRI, ultrasound is far more accessible, less expensive, portable, and easier to implement in standard operating theatres without the need for specialized equipment or infrastructure. This makes it particularly valuable in resource-limited environments or for smaller institutions without access to high-field iMRI [15]. However, limitations do exist: image quality can be affected by operator dependence, artefacts from surgical manipulation, and difficulty in distinguishing tumour from edoema or infiltrated brain in some cases [16]. Nonetheless, with growing surgeon experience and improved software, studies have shown that iUS can significantly improve the extent of resection, particularly when combined with other techniques such as fluorescence-guided surgery or intraoperative neuronavigation. However, this is supported by a few prospective studies, retrospective research and case reports. There is a need for further evaluation of the efficacy of iUS in randomized, prospective studies [17]. Nonetheless, its versatility, real-time feedback, and low cost make intraoperative ultrasound a practical and effective tool in modern neurosurgical oncology.

CEUS represents a significant advancement in intraoperative ultrasound by improving the visualization of tumour margins, vascularization, and perfusion patterns. In CEUS, microbubble contrast agents—typically composed of gas-filled lipid or protein shells—are intravenously injected and remain intravascular, enhancing the echogenicity of blood flow without crossing the blood–brain barrier [18]. This property allows for high-resolution, real-time mapping of tumour vascularity and helps differentiate viable tumour tissue from necrotic regions or surrounding edoema. CEUS is especially useful in the resection of HGG and metastases, which often exhibit abnormal or increased vascularity compared to normal brain tissue [13]. Surgeons can use this technique intraoperatively to delineate tumour boundaries better, or to identify residual tumour that may not be visible on conventional B-mode ultrasound, thus reducing the likelihood of leaving it behind. Moreover, CEUS provides rapid feedback without significantly interrupting the surgical workflow, making it a practical addition to operating rooms already equipped for standard ultrasound imaging.

One of the emerging uses of CEUS is its combination with navigated 3D ultrasound, allowing enhanced vascular imaging to be integrated into real-time surgical navigation systems. Additionally, studies suggest that CEUS may help evaluate treatment response in recurrent tumours by detecting changes in their perfusion or necrosis, although more research is needed to validate these applications in clinical practice. Importantly, CEUS is safe, with a low risk of allergic reaction or adverse effects, and does not involve ionizing radiation, which is especially advantageous in pediatric or repeat-surgery populations. Despite some limitations—such as the need for experience in interpreting dynamic perfusion images and potential variability in enhancement patterns—CEUS has demonstrated strong potential in enhancing the surgeon’s ability to perform complete, safe tumour resection.

## 3. Intraoperative Magnetic Resonance Imaging (iMRI)

The use of iMRI during brain tumour surgery has revolutionized neurosurgical oncology by significantly improving the extent of tumour resection while preserving neurological function. Brain tumours, particularly HGG, are notoriously infiltrative, and achieving GTR is a critical goal to prolong survival and reduce recurrence rate [19]. However, traditional approaches that rely solely on preoperative imaging face significant limitations due to intraoperative brain shift, which occurs as a result of cerebrospinal fluid loss, tumour debulking and changes in brain morphology during surgery [20]. This shift can displace the anatomical structures by several millimetres, rendering preoperative navigation systems inaccurate mid-procedure. Intraoperative MRI addresses this challenge by providing real-time, high-resolution images of the surgical field, allowing for dynamic updates to navigation systems and re-evaluation of the extent of resection.

In glioma surgery, studies have demonstrated that iMRI increases the rate of GTR, a key prognostic factor in patient outcomes. A randomized controlled trial by Senft et al. (2011) showed that GTR was achieved in 96% of patients in the iMRI group versus only 68% in the control group without iMRI, without a significant increase in new neurological deficits [21]. This highlights the technique’s value not only in improving oncological outcomes but also in preserving neurological function. Additionally, iMRI is capable of detecting small tumour remnants that are not visible through the operating microscope or neuronavigation based on static imaging, particularly when tumour margins are close to eloquent brain areas [22].

Different types of iMRI systems are available, including low-field (0.15–0.5 Tesla) and high-field (1.5–3.0 Tesla) magnets. High-field systems provide superior image quality and are more effective in detecting small residual tumours, but they also require more complex infrastructure, such as specialized operating suites with magnetic shielding and MRI-compatible surgical tools [23]. Despite these logistical and financial challenges, the clinical benefits of iMRI have led to its adoption in many high-volume neurosurgical centres. Furthermore, iMRI can be integrated with other technologies such as intraoperative neurophysiological monitoring, fluorescence-guided surgery (e.g., 5-ALA), and awake craniotomy protocols, creating a multimodal approach that enhances both safety and efficacy [24].

From a health economics perspective, while the upfront costs of iMRI systems and longer operative times are considerable, they may be offset by reduced rates of tumour recurrence, fewer revision surgeries, and improved quality of life [25]. In pediatric neuro-oncology, where preservation of cognitive function is critical, iMRI is especially valuable in balancing maximal tumour removal with minimal neurological damage. As imaging technology and MRI-compatible tools continue to evolve, future directions for iMRI include its integration with real-time functional imaging and artificial intelligence to further personalize and optimize surgical strategies [26].

## 4. Intraoperative Computed Tomography (iCT)

Intraoperative computed tomography has emerged as a useful imaging modality in neurosurgery, particularly for verifying the extent of tumour resection and detecting complications such as hemorrhage or brain shift in real-time. It provides fast, repeatable imaging during surgery and can be seamlessly integrated with modern neuronavigation systems to update anatomical data as the surgery progresses. Intraoperative CT is especially valuable in procedures where precision is essential—such as skull base surgeries, endoscopic approaches, and frameless stereotactic biopsies—because it can confirm anatomical accuracy, implant placement, and tissue removal without the delays associated with iMRI [27]. In terms of workflow efficiency, iCT can be performed quickly and conveniently within the operating room, reducing the need to move the patient or interrupt the surgical procedure for long periods. It is also less expensive and compatible with standard surgical instruments compared to MRI, making it more accessible for institutions without high-field iMRI systems [28,29].

However, iCT also has significant limitations that affect its utility in neuro-oncological surgery. One of the main drawbacks is its lower soft tissue contrast resolution compared to MRI, making it less effective in visualizing tumour margins, brain edoema, and subtle infiltrative tumour tissue—particularly in gliomas or metastases with ill-defined borders [30,31]. As a result, iCT is generally insufficient for guiding the resection of infiltrative brain tumours when used alone. Additionally, iCT is highly prone to artefacts, including beam hardening, metal artefacts from surgical tools, and motion artefacts, which can degrade image quality and reduce diagnostic accuracy [32,33]. Another limitation is radiation exposure, which, while minimal with modern low-dose protocols, remains a concern—especially for pediatric patients or those undergoing multiple scans during surgery. Moreover, the rigid structures of intraoperative CT systems can impose spatial constraints within the operating theatre, particularly in smaller or non-hybrid environments. Finally, while iCT can be fused with preoperative MRI to create a “virtual MRI” using deformable image registration software (such as Brainlab’s vMRI technology), this process is dependent on image quality, precise calibration, and remains unproven in terms of clinical outcome benefits in randomized trials [34].

Despite these limitations, iCT remains a practical and efficient tool for certain neurosurgical procedures where bone, calcification, or ventricular anatomy must be clearly visualized. However, its role in tumour resections—particularly for infiltrative or non-contrast enhancing lesions—remains secondary to intraoperative MRI or ultrasound, which offer superior soft tissue characterization and real-time adaptability.

## 5. Virtual MRI (vMRI)

Virtual MRI, often referred to as flexible fusion imaging, is an innovative imaging solution that aims to combine the speed of iCT with the diagnostic accuracy of preoperative MRI. This method is particularly relevant in neurosurgical procedures where real-time anatomical feedback is crucial, such as tumour resection. One of the leading platforms implementing this technology is Brainlab’s Elements Virtual iMRI software, which allows for the fusion of iCT scans with preoperative high-resolution MRI to generate an updated, MRI-like visualization without the logistical complexity and cost associated with full iMRI [35].

The core advantage of this technology lies in its ability to dynamically adjust for intraoperative changes, including brain shift, by registering the updated anatomy captured with iCT to the detailed preoperative MR data. Using advanced deformable image registration algorithms, the software recalculates and maps MR datasets to match the patient’s current anatomy seen on CT, resulting in a vMRI image that more accurately reflects intraoperative conditions [35]. This enables the surgical team to verify the extent of resection, visualize a residual tumour, and improve spatial awareness during critical phases of surgery without interrupting the workflow for long imaging sessions or moving the patient to an MRI suite.

Unlike traditional neuronavigation, which can become less reliable as the brain shifts during surgery, vMRI provides a continuously updated visual map that retains the high contrast and soft-tissue detail of MRI with the real-time adaptability of CT. Moreover, vMRI is highly integrated into navigation systems like Brainlab Curve or Cranial Navigation, meaning the fused images can be immediately used for intraoperative guidance, re-planning, and even overlay onto microscope views in real time [36].

However, despite its promise, vMRI remains largely unvalidated in clinical trials, and there is currently no evidence-based medicine (EBM) proof of its efficacy in improving clinical outcomes or resection completeness [37]. While early clinical experiences and technical evaluations suggest potential benefits, randomized controlled studies are needed to confirm whether flexible fusion imaging offers significant advantages over either iCT or iMRI alone. Additionally, while it avoids some of the high infrastructural demands of iMRI, it still requires compatible navigation software, a calibrated CT scanner, and operator familiarity with registration and image quality assessment.

In summary, vMRI platforms such as Brainlab’s vMRI offer a compelling bridge between real-time adaptability and high-resolution imaging during brain surgery. While it holds great promise for improving intraoperative decision-making and surgical outcomes, further clinical validation is essential to establish its role in standard neurosurgical oncology protocols.

## 6. 5-Aminolevulinic Acid (5-ALA) Guided Surgery

5-Aminolevulinic acid is a fluorescent molecule that has become a valuable adjunct in the surgical resection of HGG, particularly GB. Administered orally approximately three hours prior to surgery, 5-ALA is metabolized within malignant cells into protoporphyrin IX (PpIX), a compound that accumulates selectively in tumour cells due to their altered mitochondrial function and increased metabolic activity [38]. Under blue-violet light PpIX fluoresces visibly, allowing the surgeon to distinguish tumour tissue from surrounding brain parenchyma in real time. This enables GTR while minimizing damage to healthy, functional brain regions [38].

The effectiveness of 5-ALA is supported by EBM. A pivotal randomized controlled trial (RCT) published by Stummer et al. (2006) in The Lancet Oncology demonstrated that 5-ALA-guided surgery achieved a significantly higher rate of gross total resection (65%) compared to conventional white-light surgery (36%) and led to better progression-free survival at 6 months. However, most of the studies examining the usefulness of 5-ALA are non-randomized, retrospective trials [38].

However, certain limitations exist. Firstly, the effectiveness of 5-ALA is strongly dependent on tumour histology. Low-grade gliomas (LGG) and non-glial tumours (e.g., meningiomas, lymphomas, metastases) generally do not accumulate sufficient PpIX to fluoresce reliably due to intact blood–brain barriers and different metabolic profiles [39]. Secondly, while 5-ALA allows for visible fluorescence, its interpretation is subjective and operator-dependent, which can lead to inconsistencies in defining resection margins. Thirdly, there is a lack of depth resolution—5-ALA cannot show subsurface tumours, meaning deep residual tissue may be missed unless combined with other modalities like iMRI or ultrasound. Fourthly, false positives may occur, especially in areas of inflammation or reactive gliosis, which can also emit fluorescence and lead to unnecessary tissue removal [16].

From a safety perspective, 5-ALA is generally well-tolerated, but patients must avoid direct sunlight or strong light sources for at least 24 h after surgery due to the risk of photosensitivity reactions. Additionally, it may not be suitable for patients with liver disease or porphyria. Finally, while 5-ALA adds relatively low cost compared to technologies like iMRI, it still requires trained personnel and specialized surgical microscopes with fluorescence modules [38,39].

In conclusion, 5-ALA is a validated, EBM-supported technique that improves surgical outcomes in HGG resections by increasing the extent of resection, which correlates directly with survival. However, its utility is best optimized when combined with other intraoperative tools—such as neuro-navigation, iMRI, or ultrasound—to overcome its inherent limitations. Future directions include the development of quantitative fluorescence systems and the combination of 5-ALA with other targeted fluorophores for broader oncological applications.

## 7. Sodium Fluorescein Guided Surgery

Sodium fluorescein is a fluorescent dye that has gained increasing attention as a valuable tool in intraoperative imaging during brain tumour surgery. Unlike 5-ALA, which selectively accumulates in tumour cells via metabolic pathways, fluorescein distributes passively through areas of blood–brain barrier (BBB) disruption, making it particularly effective in visualizing contrast-enhancing lesions such as HGG, metastases, and some meningiomas [16]. Administered intravenously—typically in low doses (2–5 mg/kg)—shortly before dural opening, fluorescein fluoresces when excited with yellow (560–570 nm) light, allowing surgeons to visualize tumour tissue using a surgical microscope equipped with a YELLOW 560 filter system (e.g., Zeiss Pentero or KINEVO platforms) [40].

The main advantage of fluorescein-guided surgery is its ability to highlight tumour margins intraoperatively in real time, improving the surgeon’s capacity to achieve gross GTR. Several studies, including prospective cohort trials, have demonstrated that fluorescein-guided resection is associated with higher GTR rates and improved progression-free survival, particularly in patients with HGG [40,41]. A multicentre observational study by Acerbi et al. (2018) involving over 200 patients showed that fluorescein was safe, well-tolerated, and led to complete resection in more than 80% of patients with enhancing gliomas [40]. Importantly, fluorescein is significantly less expensive than 5-ALA and does not require preoperative ingestion or special timing, making it easier to integrate into routine workflows, especially in resource-limited settings.

However, fluorescein also has important limitations. Because its mechanism of action is based on extravasation through a disrupted BBB, it cannot distinguish between tumour and other enhancing pathologies such as radiation necrosis, inflammation, or postoperative ischemia, potentially leading to false positives [42]. Furthermore, fluorescein is non-specific to tumour cells and may also accumulate in oedematous tissue or peritumoral regions, making interpretation challenging in infiltrative tumours. In LGG or tumours with intact BBB, fluorescein has limited utility as these regions may not enhance on MRI or fluoresce under surgical lighting. Additionally, although generally safe, adverse effects such as nausea, urticaria, hypotension, or rare anaphylactic reactions have been reported, especially when higher doses are used. Moreover, the technique necessitates the use of a surgical microscope with dedicated filter systems for fluorescein visualization. Although widely available in neurosurgical centres, such equipment is still lacking in some hospitals [43].

From an EBM standpoint, fluorescein is supported primarily by observational studies, case series, and prospective cohort trials, but has not yet established its efficacy to the same degree as 5-ALA. As such, while fluorescein has demonstrated encouraging results and is increasingly used in Europe and other regions, it has not yet received the same level of regulatory approval in all countries (e.g., it is not FDA-approved in the U.S. for brain tumour surgery) [44]. Nonetheless, the low cost, real-time visualization, and ease of use make fluorescein-guided surgery a promising adjunct, particularly when used alongside neuronavigation or iUS for better anatomical context. Analysis of advantages and disadvantages of fluorescein compared to other methods discussed in the article are summarised in Table 1.

## 8. Conclusions

In conclusion, modern brain tumour surgery has greatly evolved through the integration of advanced intraoperative imaging and visualization techniques aimed at achieving maximal safe resection—a key determinant of patient survival and functional outcome. Each modality offers distinct advantages and limitations, and their appropriate selection often depends on tumour type, location, institutional resources, and surgeon expertise. iMRI remains the gold standard for anatomical accuracy and detailed visualization of residual tumour, particularly in high-grade gliomas, though it requires significant infrastructure and increases operative time. iCT provides rapid imaging and is valuable for verifying bony anatomy or hemorrhage; however, its limited soft tissue resolution and susceptibility to artifacts reduce its utility in infiltrative tumours. iUS stands out for its real-time, radiation-free imaging and adaptability to brain shift, especially when enhanced with contrast agents (CEUS) or integrated with navigation systems—making it highly practical and cost-effective. vMRI, based on fusing iCT with preoperative MRI (e.g., Brainlab software), offers a promising hybrid solution with improved anatomical fidelity; however, clinical validation is still pending. Meanwhile, fluorescence-guided surgery with 5-ALA or fluorescein has revolutionized the visual identification of tumour margins: 5-ALA, supported by evidence from randomized trials, is highly effective in high-grade gliomas, while fluorescein offers a more accessible, real-time alternative based on BBB disruption, albeit with fewer large-scale studies. Ultimately, the optimal approach may lie in a multimodal strategy, combining several techniques—such as fluorescence, intraoperative imaging, and neuro-navigation—to tailor surgical planning and execution to each individual case. As technology advances and clinical research continues to expand, these tools collectively represent a new era in neurosurgical oncology, one that prioritizes precision, safety, and long-term patient outcomes.

## Figures and Tables

**Table 1 biomedicines-13-02550-t001:** Summary table.

Tool	Method	Advantages	Disadvantages	Approximate Cost
Intraoperative Ultrasound (iUS)	Provides dynamic, real-time visualization of the brain and tumour during surgery, aiding localisation, assessment of resection progress, and detection of residual tumour.	Continuous use throughout surgery without interrupting the workflow. Compensates for brain shift. Portable. Does not require a highly specialized surgical infrastructure. Not as expensive as iCT and iMRI.	Requires considerable operator experience to interpret images accurately. May be difficult to distinguish tumour from edoema or infiltrated brain tissue in some cases. supported by a few prospective studies, retrospective research and case reports.	EUR 7000–EUR 11,000
Intraoperative Magnetic Resonance Imaging (iMRI)	Utilizes MRI during surgery to provide real-time, high-resolution images of the operative field.	High-quality imaging—the gold standard for brain tumours. Compensates for brain shift. Detects even small residual tumours.	Very expensive. Requires a specially designed operating theatre and equipment. Prolonged image acquisition time interrupts surgical workflow.	Over EUR 1 million (however, costs may be offset by reduced tumour recurrence, fewer revision surgeries, and improved quality of life).
Intraoperative Computed Tomography (iCT)	Provides intraoperative imaging of the surgical field, often used with neuronavigation.	Relatively fast image acquisition. Particularly useful in skull base procedures, endoscopic approaches, and frameless stereotactic biopsies. Compensates for brain shift.	Requires specialized equipment. Lower soft-tissue resolution than MRI -insufficient for guiding resection of infiltrative tumours when used alone. Susceptible to artefacts. Radiation exposure. Occupies considerable space in the operating theatre.	EUR 80,000–EUR 300,000 (costs may be mitigated by reduced recurrence rates, fewer re-operations, and improved patient outcomes).
Virtual Magnetic Resonance Imaging (vMRI)	Combines intraoperative CT scans with preoperative high-resolution MRI to generate an updated, MRI-like visualization.	Adjusts for intraoperative anatomical changes. Integrated into neuronavigation systems.	Limited evidence supporting improved clinical outcomes or the extent of resection. Requires specialized hardware and software.	Cost equivalent to intraoperative CT plus software licencing.
5-Aminolevulinic Acid (5-ALA)	A precursor metabolized within malignant cells into protoporphyrin IX, which accumulates selectively due to altered mitochondrial function and high metabolic activity.	Enhances intraoperative visualization of tumour margins. Relatively inexpensive compared with advanced imaging systems. Supported by one randomized, prospective study, retrospective studies and case reports.	Effectiveness depends on tumour histology. Requires specific optical filters and microscopes. Interpretation is subjective. May yield false positives in inflammatory or gliotic areas.	Several thousand euros.
Fluorescein	Passively distributes through regions of blood–brain barrier disruption, highlighting contrast-enhancing areas.	Provides real-time intraoperative visualization of tumour margins.	Non-specific -cannot reliably differentiate tumour tissue from other enhancing pathologies (e.g., radiation necrosis, inflammation, ischaemia). Possible adverse effects include nausea, urticaria, hypotension, or rare anaphylaxis. Requires a microscope with dedicated fluorescence filters. Supported mainly by observational studies and case reports.	EUR 30–EUR 50 per dose (excluding equipment cost).

## Data Availability

Not applicable.

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
