# Peer review of "Advancing Precision in Neuro-Oncology with Intraoperative Imaging and Fluorescence Guidance: A Narrative Review"

_biomedicines, 2025, doi:10.3390/biomedicines13102550_

Round 1

Reviewer 1 Report

Comments and Suggestions for Authors

The article provides a well-structured and up-to-date overview of the main intraoperative techniques in neuro-oncology (iMRI, iCT, iUS, vMRI, 5-ALA, and fluorescein). It balances technical explanation, clinical evidence, and future perspectives, highlighting both advantages and limitations of each approach.The conclusion is clear, emphasizing the importance of multimodal strategies and the need to adapt technology choices to institutional resources and clinical context.

on the other hands, after more points need improvement:

  • Although presented as a “narrative review,” the paper lacks methodological rigor (no clear inclusion/exclusion criteria, no systematic evaluation of evidence quality). This reduces its scientific weight.
  • Some sections are more descriptive than critical. For example, vMRI and fluorescein are discussed with limited depth regarding the strength of available evidence.
  • The writing is somewhat redundant; a comparative summary table could improve clarity and avoid repetition.
  • The absence of figures, diagrams, or schematics makes it harder to quickly compare techniques.
  • The discussion on health economics is underdeveloped. A comparative cost-effectiveness perspective would strengthen the practical impact of the review.
  • The English language requires revision to improve fluency and precision.
  • Including a brief discussion on the use of fluorescence in spinal pathology could broaden the scope and increase the overall impact of the article ( 

    https://doi.org/10.1016/j.wneu.2022.09.117; 10.1016/j.wneu.2017.09.061; https://doi.org/10.1016/j.wneu.2017.10.069 )

Comments on the Quality of English Language

Need a revisioni of syntax and Grammar 

Author Response

Dear Reviewer,

Thank you very much for your insightful and constructive comments. We have carefully considered each point and revised the manuscript accordingly. Our detailed responses are provided below.

Comment: Although presented as a “narrative review,” the paper lacks methodological rigour (no clear inclusion/exclusion criteria, no systematic evaluation of evidence quality). This reduces its scientific weight.
Response:
Thank you for this valuable remark. We deliberately chose the narrative review format to highlight the most important aspects of the evolving armamentarium of intraoperative technologies shaping the future of precision neuro-oncology. This approach allows for a broader exploration of the literature, enabling inclusion of clinically relevant studies and expert perspectives while emphasising current trends and key developments from the standpoint of experienced neurosurgeons.

Comment: Some sections are more descriptive than critical. For example, vMRI and fluorescein are discussed with limited depth regarding the strength of available evidence.
Response:
Thank you for this helpful comment. We have revised these sections to provide greater depth and critical appraisal of the available evidence. Key findings and comparative data from the literature are now summarised in the accompanying table for clarity and completeness.

Comment: The writing is somewhat redundant; a comparative summary table could improve clarity and avoid repetition.
Response:
Thank you for this remark. We have added a comprehensive summary table that concisely presents comparative information on the discussed techniques, thereby improving clarity and reducing redundancy.

Comment: The absence of figures, diagrams, or schematics makes it harder to quickly compare techniques.
Response:
Thank you for this important observation. We have included a detailed summary table to facilitate comparison of the various intraoperative imaging and visualisation techniques.

Comment: The discussion on health economics is underdeveloped. A comparative cost-effectiveness perspective would strengthen the practical impact of the review.
Response:
Thank you for this insightful suggestion. We have now incorporated a comparison of health economics within the summary table, including indicative cost ranges for each technique, to enhance the practical relevance of the review.

Comment: Including a brief discussion on the use of fluorescence in spinal pathology could broaden the scope and increase the overall impact of the article (e.g. https://doi.org/10.1016/j.wneu.2022.09.117; 10.1016/j.wneu.2017.09.061; https://doi.org/10.1016/j.wneu.2017.10.069).
Response:
Thank you for this remark. As the primary focus of this narrative review is on high-grade gliomas, we decided to omit spinal applications. While the topic of fluorescence in spinal pathology is indeed of interest, it represents a distinct and extensive area that would not be adequately addressed within the current manuscript’s scope.

Reviewer 2 Report

Comments and Suggestions for Authors

The manuscript reviews methods used by neurosurgeons to improve dissecting the neoplasms by surgery.  However, some points need to be clarified:

  1. As the authors despite, the gold standard use neuroimaging studies to delineate the tumor (P4, 5, 6).
  2. Fluoresein injection is only used as a dye enhancement in cases with BBB disruption; limitations are several, because fluorescent dyes are useful when coupled with antibodies for histological studies using microscopes with specific light filters for fluorescence. The authors describe the use of fluorescein injections which would require specific light sources to guide the surgical procedure, this technique does require specific equipment (P7,L307-316) which does limit their use, and it has not been approved in several countries. Although the authors describe fluorescence guidance as a valuable tool and not very expensive (P7, L324-325) it does need special equipment to be utilized thus raising costs, other dyes like Blue Evans could be studied for delineation of BBB disruption without the implementation of complex technologies.
  3. The title indicates a “narrative” review which should be described briefly in the text why this is a “narrative review” and not a “review”.
  4. A neoplasm that represents 4% of a given tumor is not “rare” (P2,L54) but “less frequent”.
  5. International experience and number of reports indicate that imaging studies, rather than fluorescent guidance, are the intraoperative tools more used by neuro-oncologists during surgical extirpation.

Author Response

Dear Reviewer,

Thank you very much for your insightful and constructive comments. We have carefully considered each point and revised the manuscript accordingly. Our detailed responses are provided below.

Comment #1:

As the authors describe, the gold standard is the use of neuroimaging studies to delineate the tumour (P4, 5, 6).

Response:

We have revised the paragraph concerning the use of MRI in brain tumours as follows:

“Magnetic resonance imaging (MRI) is the gold standard for the preoperative diagnosis of brain tumours. Imaging sequences such as T1, T2, FLAIR, SWI, DWI, and tractography provide information about tumour cellularity, cerebral oedema, invasion, and white matter tracts. High-grade tumours typically demonstrate strong contrast enhancement and frequent central necrosis, resulting in a ring-enhancing appearance. Functional MRI enables mapping of eloquent cortical centres, their relationship to the tumour, and associated neuroplasticity.” (Lines 69–74)

Comment #2:

Fluorescein injection is used as a dye enhancement in cases with BBB disruption. There are several limitations, as fluorescent dyes are mainly useful when coupled with antibodies for histological studies using microscopes with fluorescence filters. The authors describe the use of fluorescein injections, which require specific light sources to guide the surgical procedure. This technique demands special equipment (P7, L307–316), limiting its use, and it has not been approved in several countries. Although the authors describe fluorescence guidance as a valuable and inexpensive tool (P7, L324–325), it does require specialised equipment, thus increasing costs. Other dyes, such as Evans Blue, could be studied for delineation of BBB disruption without the need for complex technologies.

Response:

Thank you for this valuable remark. We have now addressed this point by including a statement regarding the need for specialised equipment in the paragraph on the disadvantages:

“Moreover, the technique necessitates the use of a surgical microscope with dedicated filter systems for fluorescein visualisation. Although widely available in neurosurgical centres, such equipment is still lacking in some hospitals.” (Lines 341–344)

Comment #3:

The title indicates a “narrative review”, which should be briefly described in the text to explain why this is a narrative review rather than a general review.

Response:

We appreciate this suggestion. We have added an explanatory statement to the introduction:

“The narrative review format was chosen to highlight the most important aspects of the evolving armamentarium of intraoperative technologies shaping the future of precision neuro-oncology.”

Comment #4:

A neoplasm that represents 4% of a given tumour type is not ‘rare’ (P2, L54) but rather ‘less frequent’.

Response:

Thank you for this observation. We have revised the sentence accordingly to replace “rare” with “less frequent.”

Comment #5:

International experience and the number of reports indicate that imaging studies, rather than fluorescence guidance, are the intraoperative tools most frequently used by neuro-oncologists during surgical extirpation.

Response:

Thank you for this important remark. We have emphasised this in the introduction:

“To aid resection, surgeons most commonly use intraoperative ultrasound (iUS), computed tomography (iCT), or MRI. Other tools include 5-aminolevulinic acid (5-ALA), which improves intraoperative identification of the tumour by visualising it under ultraviolet light, and fluorescein, which accumulates in areas of blood-brain barrier disruption, aiding visualisation of contrast-enhancing lesions.” (Lines 89–93)

Reviewer 3 Report

Comments and Suggestions for Authors

The review article "Advancing Precision in Neuro-Oncology with Intraoperative Imaging and Fluorescence Guidance: A Narrative Review" is devoted to imaging and visualization techniques for surgery of various tumors of the central nervous system and brain.

The article has a number of disadvantages:

  1. The introduction section in the review article is too general, the authors paid a lot of attention to the description of glioblastoma and astrocytoma, the course of the disease, and the symptoms. At the same time, little attention is paid to the complexities of visualizing these tumors and why all these imaging techniques are needed for them at all.
  2. There is not a single picture in the review that would facilitate the perception of the material and would allow it to be summarized. In addition, there is no summary table that provides brief information on each of the described methods.

Otherwise, the authors briefly list the main existing approaches to intraoperative imaging of tumors, not in toomuch detail, but enough. I recommend that this review be published after major revision, which is necessary toimprove the content of this review.

Author Response

Dear Reviewer,

Thank you for the insightful comments.

Comment #1: The introduction section in the review article is too general, the authors paid a lot of attention to the description of glioblastoma and astrocytoma, the course of the disease, and the symptoms. At the same time, little attention is paid to the complexities of visualizing these tumors and why all these imaging techniques are needed for them at all.

Response: Thank you for this comment. We revised the introduction accordingly.

Comment #2: There is not a single picture in the review that would facilitate the perception of the material and would allow it to be summarized. In addition, there is no summary table that provides brief information on each of the described methods.

Response: Thank you for this remark. We included a table in the manuscript.

Round 2

Reviewer 1 Report

Comments and Suggestions for Authors

good improvmentes 

Reviewer 3 Report

Comments and Suggestions for Authors

The authors took into account all the comments and made necessary changes. The article can be published in its current form.